# Bio-Based Polymer Developments from Tall Oil Fatty Acids by Exploiting Michael Addition

**DOI:** 10.3390/polym14194068

**Published:** 2022-09-28

**Authors:** Ralfs Pomilovskis, Inese Mierina, Anda Fridrihsone, Mikelis Kirpluks

**Affiliations:** 1Polymer Laboratory, Latvian State Institute of Wood Chemistry, Dzerbenes St. 27, LV-1006 Riga, Latvia; 2Institute of Technology of Organic Chemistry, Faculty of Materials Science and Applied Chemistry, Riga Technical University, P. Valdena St. 3/7, LV-1048 Riga, Latvia

**Keywords:** fatty acid-based Michael donor, Michael addition, bio-based polymer, tall oil

## Abstract

In this study, previously developed acetoacetates of two tall-oil-based and two commercial polyols were used to obtain polymers by the Michael reaction. The development of polymer formulations with varying cross-link density was enabled by different bio-based monomers in combination with different acrylates—bisphenol A ethoxylate diacrylate, trimethylolpropane triacrylate, and pentaerythritol tetraacrylate. New polymer materials are based on the same polyols that are suitable for polyurethanes. The new polymers have qualities comparable to polyurethanes and are obtained without the drawbacks that come with polyurethane extractions, such as the use of hazardous isocyanates or reactions under harsh conditions in the case of non-isocyanate polyurethanes. Dynamic mechanical analysis, differential scanning calorimetry, thermal gravimetric analysis, and universal strength testing equipment were used to investigate the physical and thermal characteristics of the created polymers. Polymers with a wide range of thermal and mechanical properties were obtained (glass transition temperature from 21 to 63 °C; tensile modulus (Young’s) from 8 MPa to 2710 MPa and tensile strength from 4 to 52 MPa). The synthesized polymers are thermally stable up to 300 °C. The suggested method may be used to make two-component polymer foams, coatings, resins, and composite matrices.

## 1. Introduction

The European Commission has set up a “European Green Deal” to tackle climate and environmental challenges. It is a new growth strategy that aims to transform the European Union into a prosperous society with a modern, resource-efficient, and competitive economy, achieving climate neutrality and a balance in carbon emissions and removals by 2050, which means reducing net greenhouse gas emissions to zero [1]. Improving the polymer industry by introducing innovative, sustainable, and knowledge-intensive technologies for producing materials from renewable bio-resources is also an essential field for achieving this goal [2].

In the polyurethane industry, sustainability has emerged as a major concern. Conventionally, polyurethanes are produced from two components—A (polyol) and B (isocyanate)—that are derived from petrochemicals. Particular attention is given towards the synthesis of polyols from biological fatty acids, diglycerides, and triglycerides [3,4,5,6,7,8]. Such bio-polyols are suitable for the production of polyurethanes as a method to make polyurethanes more sustainable [9]. However, the use of isocyanates still remains problematic as isocyanates are considered as toxic substances relative to the environment [10,11,12,13,14].

Recently, one of the extensive research topics is the development of non-isocyanate polyurethanes (NIPUs) materials as an alternative to polyurethanes. This technology has significant drawbacks, such as high pressure, high temperature, and long reaction times [15,16]. The polymer industry requires alternative methods for producing polymers in relatively mild conditions with similar properties to polyurethanes. The Michael reaction can be used as a new synthesis pathway to yield bio-based polymers with a wide range of properties [2,17].

The Michael reaction is a versatile synthesis type for linking electron-poor olefins to a wide range of nucleophiles [18]. Enolates [19,20], nitrogen-containing [21,22,23,24,25,26] reagents, and thiols [26,27,28,29,30,31,32] are typical Michael donors, while alkyl acrylates are mainly used as Michael acceptors [33]. Adding an enolate anion to the α,β-unsaturated carbonyl compound to form a carbon–carbon bond is a Michael reaction [19,20,34]. A type of polymerization for the addition of hydroxyl monomers with electrophilic double or triple bonds is called the oxa-Michael reaction [35,36]. In the case of nitrogen and sulphur-containing donors, the reaction is called aza-Michael [21,22,23,24,25,26,37] and thio-Michael [26,27,28,29,30,31,32] addition reactions, respectively. In recent years, aza- and thio-Michael reactions have gained attention for polymerization [35,38]. However, carbon-Michael reactions have gained less attention.

The majority of the studies has been conducted on coating materials [39,40,41,42,43]. Recently, Naga et al. presented Michael-addition reactions of multi-functional acetoacetate and diacrylate monomers, yielding corresponding gels at room temperature in the presence of 1,8-diazabicyclo[5.5.0]undecane-7-ene as a catalyst [44]. Park reported how hydrophobic thin-film nanocomposite membranes could be successfully developed using only naturally occurring compounds by interfacial polymerization [45]. Sinha et al. has concluded that the nucleophilicity of the used enamines had a significant impact on the efficiency and total reaction yields. Moreover, photo-protecting amines with 2-(2-nitrophenyl)propyl chloroformate in the presence of a ketone create enamines in situ [46].

The Michael reaction can yield a variety of linear, graft, dendric, highly cross-linked, and hyperbranched polymer networks [18,33]. The mechanism of the reaction is relatively well-studied and straightforward. The thermodynamical mechanism depends on the base’s relative strengths and the type of acetoacetate. The base deprotonates the acetoacetate, resulting in an enolate anion. The enolate anion then reacts with the acrylate in a 1,4-conjugate addition, and the carbonyl of the acrylate stabilizes, allowing the base to regenerate (see Figure 1). The enthalpic transition that occurs when a π-bond is replaced with a σ-bond is the overall driving force for conjugate addition. As a result, 1,4-addition is preferred over 1,2-addition. The product of the first Michael reaction contains active methylene hydrogen that can be added to another acrylate in a second step [14,34], which means that the Michael donor reacts twice with two Michael acceptors [33], of course, if the ratio allows it. The polymeric material can only be formed if the molecule of the Michael acceptor component contains at least two functional groups.

The reaction between acrylates and the active methylene group of β-ketoesters is quick, and the system cures at room temperature after mixing it in the presence of catalysts. This reaction is a strong base-catalyzed addition [18,20,34]. For the Michael reaction amine, amidine- and guanidine-based catalysts may be the most suitable [47,48] for example 1,1,3,3-tetramethylguanidine, triethylamine, 1,4-diazabicyclo[2.2.2]octane, (1,8-diazabicyclo[5.4.0]undec-7-ene), 4,4′-methylenebiscycloheexanamine, (1,5-diazabicyclo[4.3.0]non-5-ene), and 7-methyl-1,5,7-triazabicyclo[0,4,4]dec-5-ene [18,20,34,39,42,48,49,50,51].

In this study, we used previously developed acetoacetates of two tall-oil-based and two commercial polyols to obtain polymers by the Michael reaction. The previously developed acetoacetates had different chemical structures and functionalities. In conjunction with the use of various acrylates in polymerization reactions, it allowed the development of polymer formulations with varied cross-link density. Three acrylates with different functionalities—bisphenol A ethoxylate diacrylate (BPAEDA), trimethylolpropane triacrylate (TMPTA), and pentaerythritol tetraacrylate (PETA)—were used. The physical and thermal properties of the developed polymers were studied using dynamic mechanical analysis (DMA), differential scanning calorimetry (DSC), thermal gravimetric analysis (TGA), and a universal strength testing machine. The chemical structure and functional groups present in the developed polymer were determined using Fourier-transform infrared spectroscopy (FTIR). The proposed technology has a potential in the production of two-component polymer foams, coatings, resins, and matrix for composites.

## 2. Materials and Methods

### 2.1. Materials

Epoxidized tall oil fatty acids 1,4-butanediol polyol acetoacetate (E^IR^TOFA_DB_AA) were obtained from polyol that was synthesized from tall oil fatty acids via epoxidation using ion exchange resin (E^IR^TOFA) followed by oxirane ring-opening and esterification with 1,4-butandiol and subsequent acetoacetylation with *tert*-butyl acetoacetate by a transesterification reaction, acid value < 5 mg KOH·g^−1^, hydroxyl value 36.2 mg KOH·g^−1^, moisture 0.025%, and acetoacetate groups 0.3307 mol·100 g^−1^. Epoxidized tall oil fatty acids trimethylolpropane polyol acetoacetate (E^IR^TOFA_TMP_AA) was obtained from E^IR^TOFA by oxirane ring-opening and esterification with trimethylolpropane and subsequent acetoacetylation with *tert*-butyl acetoacetate by a transesterification reaction, acid value < 5 mg KOH·g^−1^, hydroxyl value 41.6 mg KOH·g^−1^, moisture 0.037%, and acetoacetate groups 0.4562 mol·100 g^−1^. Acetoacetyled Neopolyol 380 (NEO380_AA) was synthesized from Neopolyol 380 (NEO380) by transesterification with *tert*-butyl acetoacetate from the NEO Group, acid value < 5 mg KOH·g^−1^, hydroxyl value 40.7 mg KOH·g^−1^, moisture 0.048%, and acetoacetate groups 0.4242 mol·100 g^−1^. Acetoacetyled Lupranol 3300 (L3300_AA) was synthesized from Lupranol 3300 (L3300) by transesterification with *tert*-butyl acetoacetate, from BASF, acid value < 5 mg KOH·g^−1^, hydroxyl value 26.2 mg KOH·g^−1^, and moisture 0.021%, acetoacetate groups 0.4456 mol·100 g^−1^. BPAEDA, average M_n_~512, contains 1000 ppm monomethyl ether hydroquinone as an inhibitor from Sigma-Aldrich; TMPTA contains 600 ppm monomethyl ether hydroquinone as an inhibitor and is of technical grade from Sigma-Aldrich; PETA contains 350 ppm monomethyl ether hydroquinone as an inhibitor and is technical grade from Sigma-Aldrich (St. Louis, MO, USA). 1,1,3,3-tetramethylguanidine (TMG), assay 99%, was purchased from Sigma-Aldrich. Catalyst and acrylates were directly used as delivered without further purification.

### 2.2. Solid Polymer Development

The polymerization was carried out by mixing the acrylate with the catalyst (TMG), adding acetoacetate, and stirring the mixture rapidly until a homogeneous medium was obtained. Mixing was about 30 s. The ratio of acetoacetate to acrylic groups was 1:2 mol. The amount of TMG was 1% of the total mass of the mixture. After mixing, the mixture was quickly poured into a 15 mL centrifuge tube and centrifuged for 1 min to remove air bubbles and obtain a homogeneous polymer monolith. Samples were made with a weight of 14 g and solidified in a tube at room temperature and atmospheric pressure for 2 min to 3 min. Polymer syntheses were repeated twice. Samples were tested from both syntheses.

Tensile samples were prepared by immediately pouring the polymerization mixture into a mold after centrifugation. Samples were tested after 24 h curing at room temperature.

### 2.3. Characterization Methods

**FTIR.** Polymer chemical structure was analyzed using FTIR data, which was obtained with a Thermo Scientific Nicolet iS50 spectrometer (Thermo Fisher Scientific, Waltham, MA, USA) at a resolution of 4 cm^−1^ (32 scans). FTIR data were collected using the attenuated total reflectance technique with diamond crystals. A sheet of the obtained polymer sample was pressed against the prism and analyzed.

**DSC.** Metler Toledo DSC 823^e^ (Mettler Toledo, Greifensee, Switzerland) was used to obtain DSC data. STAR^e^ Software ver. 9.00 (Mettler Toledo, Greifensee, Switzerland) and OriginPro 2021 9.8.0.200 programs (Northampton, MA, USA) were used for data processing. Specimens weighing approximately 0.2–2.0 mg were cut from the polymer sample. A total of 5 mg of samples were placed in an aluminium sample pan with an accuracy of ±0.5 mg. A Mettler Toledo sealing press for crucibles was used to crimp covers on DSC pans of aluminium. The test was organized in two cycles. The start temperature of the first cycle was 25 °C with heating up to 180 °C, followed by cooling to −50 °C. The second heating cycle was from −50 °C to 180 °C, and the temperature was reduced to 25 °C in the second cooling cycle. The heating rate used was 10 °C·min^−1^. Two samples from each series were tested for DSC.

**TGA.** Thermogravimetric data were obtained using Discovery TGA equipment (TA instruments, New Castle, DE, USA). Data processing was performed using the OriginPro 2021 9.8.0.200 and TA Instruments TRIOS version #5.0.0.44608 software. A 10 mg sample consisting of similar size pieces (0.5–2.0 mg) was placed on a platinum scale sample pan with an accuracy of ±0.5 mg. Then, the sample on the sample pan was heated in a nitrogen atmosphere at 10 °C·min^−1^ in a temperature range between 30 °C and 700 °C. Two samples from each series were tested for TGA.

**DMA.** DMA was carried out with Mettler Toledo DMA/SDTA861^e^ (Mettler Toledo, Switzerland). The temperature range was from −100 °C to 180 °C, the ramp rate was 3 °C·min^−1^, the frequency was 1 Hz, amplitude was 30 μm, and maximal force was 5 N. The compression oscillation mode was used. Polymer monolith samples with a diameter of 13 mm and a height of 7 mm, with an accuracy of ±0.2 mm, were used for DMA tests. Two samples from each series were tested for DMA.

The cross-link density was calculated from the data obtained in the DMA test according to Equation (1):

(1)$$\nu_{e}=\frac{E^{\prime }}{3RT}$$
where *ν_e_* is the cross-link density (mol·cm^−3^), *E*′ is the tensile storage modulus (MPa), *R* is the gas constant (8.314 m^3^·Pa·K^−1^·mol^−1^), and *T* is the temperature corresponding to the storage modulus value into the rubbery plateau (K) [52,53,54,55].

Knowing density *ρ* (g·cm^−3^), the molecular weight between cross-links (*M_c_* (g·mol^−1^)) was calculated according to Equation (2) [56].



(2)
$$M_{c}=\frac{3RT\rho }{E_{rubbery}^{\prime }}$$



Tensile modulus (Young’s) and compressive strength were determined from tests carried out at room temperature (22 °C) using a universal machine, an ElectroPuls E3000 from Instron (Norwood, Norfolk, MA, USA), following the ISO 14125:1998 standard. The sample size was 2.5 mm × 4 mm × 25 mm. The deformation rate for all samples was 2 mm·min^−1^, and at least four samples from each series were tested.

## 3. Results and Discussion

A schematic overview of polymer syntheses performed, including information on the synthesis steps of the Michael donor monomers, is provided in Figure 2. Previously developed acetoacetates of two tall oil-based polyols (E^IR^TOFA_DB_AA and E^IR^TOFA_TMP_AA) and two commercial polyols (NEO380_AA and L3300_AA) were used to obtain polymers by the Michael reaction with three acrylates with different functionalities—BPAEDA, TMPTA, and PETA. Monomers of different degrees of functionalities were used to demonstrate that it was possible to obtain a polymeric material from tall oil polyol acetoacetates with various acrylates. It also allowed the investigation of the effect of acrylate functionality on the properties of the obtained polymeric materials.

### 3.1. FTIR Analysis of Monoliths

The FTIR spectra of the film of obtained thin polymeric materials are shown in Figure 3. The FTIR spectra did not show very intensive peaks at ~1630 cm^−1^ and ~810 cm^−1^, which are characteristic of the absorbance of acrylic group vibration. This indicated that the content of free acrylic groups in polymers was low [57]. Thus, a relatively high conversion occurred.

Polymers derived from BPAEDA have two intense peaks (at 830 cm^−1^ and 1510 cm^−1^) corresponding to aromatic bonds characterizing bisphenol A in the FTIR spectrum. In the case of NEO380_AA-based polymers, an intense peak at 730 cm^−1^ was observed in the FTIR spectra, indicating the out-of-plane C-H bending of an aromatic ring (Figure 3c).

### 3.2. DSC Analysis of Monoliths

DSC heating curves are shown in Figure 4. The glass transition temperature can be very clearly identified in almost all DSC curves of the new polymeric materials. The exception was in the case of polymers from PETA acrylate. The glass transition temperature was less pronounced in the curves of these polymers by the used DSC analysis mode. For polymers synthesized using BPAEDA, the glass transition was clearly visible. The enthalpy relaxation peak also accompanied it.

For polymers obtained from commercial polyol acetoacetate (L3300_AA and NEO380_AA), the glass transition temperature was mainly influenced by the used acrylate. The glass transition temperature differed by less than 2 °C when polymers obtained with the same acrylates were compared (Figure 4c,d). Depending on the used acrylate, the glass transition temperature increased as follows: BPAEDA < TMPTA < PETA. The functionality of the used acrylate impacted the glass’s transition temperature. When acrylate with higher functionality was used to obtain polymers, the polymer exhibited higher glass transition temperature due to a higher cross-link density in the polymer matrix [58].

The same trend was observed for polymers obtained from acetoacetylated tall oil polyols (Figure 4a,b). The used tall oil polyol acetoacetate had a significant effect on the glass transition temperature of the obtained polymer. Higher glass transition temperature was observed for E^IR^TOFA_TMP_AA, which was synthesized using a higher functionality polyol.

In the case of commercial polyol-based acetoacetate polymers, the difference in glass transition temperatures was relatively small when considering the effect of the polyol used. Of course, the previously established correlation between acrylate functionality and the glass transition temperature remained. A higher glass transition temperature was observed for polymers in which acrylate with higher functionalities was used (Figure 4c,d).

### 3.3. TGA Analysis of Monoliths

TGA curves and their derivatives of polymers obtained from tall oil-based and commercial polyol acetoacetates are shown in Figure 5. As shown in Figure 5a,b, no significant weight loss was observed up to almost 300 °C for polymers derived from tall oil polyols; they were thermally stable. This is an advantage because, for example, classic polyurethane materials without special thermal stabilizer additives decompose in the temperature range from 200 °C to 300 °C [59,60]. Moreover, organic glass poly(methyl methacrylate) (PMMA) onset decomposition temperatures are under 300 °C [61,62]. In Table 1, data of temperatures are summarized representing the temperatures at which the following mass losses were achieved: 5% (T_m5%_), 10% (T_m10%_), 25% (T_m25%_), and 50% (T_m50%_). If the influence of used polyol on thermal properties of polymer material was compared, then polymers derived from TMP polyols had higher thermal stability.

On the other hand, if polymer materials were compared according to the used acrylate, then the most thermally stable polymers were obtained from BPAEDA acrylate. This could be explained by the aromatic structure of bisphenol A. It was observed that the temperature at which a 5% mass loss occurred was lower when acrylate with higher functionality was used.

The highest T_m5%_ was observed for polymers derived using difunctional BPAEDA, followed by trifunctional TMPTA. The lowest T_m5%_ was for polymers derived from tetrafunctional PETA. Thermal degradation continued up to almost 500 °C for all samples, after which no significant weight loss was observed.

The peaks indicate the point of the greatest rate of change on the weight loss curve (Figure 5). Two distinct peaks were observed for polymers obtained using BPAEDA as the acrylate. For other polymers, the similarity of the curves was determined by the used polyol and less by the acrylate.

Polymers from commercial polyols also showed similar characteristics in TGA mass-loss curves and their derivative curves (Figure 5c,d). Polymeric materials derived from tall oil polyols with BPAEDA acrylate had higher thermal stability (when comparing the temperature for weight loss of 5%) than commercial polyols with the same acrylate. The temperature depends significantly on the used acrylate for a weight loss of 5%, but this effect was less pronounced for weight losses at higher temperatures.

### 3.4. DMA Analysis of Monoliths

DMA test was performed to determine the dynamic mechanical properties of polymer materials. The obtained curves are shown in Figure 6. Based on the tanδ data, all samples had a quite homogeneous network structure because there was only one relatively narrow peak with a clear maximum, which appears in the graphs. These relationships between peak width, symmetry and structure, and the homogeneity of the structure have been described in several studies [20,63,64,65].

Wider peaks and with a lower maximum of tanδ value were observed in the case of the tall oil acetoacetate polymer with a tetrafunctional acrylate ester PETA (Figure 6a,b). This indicated a lower level of homogeneity in cross-link density. A symmetrical peak with the highest tanδ value was reached for all polymer samples obtained using the difunctional acrylate BPAEDA, indicating higher homogeneity of the cross-linked network.

The glass transition temperatures determined by the DMA method are summarized in Table 2. The glass transition temperatures obtained from DMA were compared with the glass transition temperatures of materials from DSC analysis. Although the glass transition temperatures were different, same trends between the glass transition temperatures were observed. The glass transition temperatures determined using the DMA method were higher than those determined using DSC analysis. DMA method has a higher sensitivity than DSC for detecting the glass transition temperature [66]. DMA is beneficial for polymers with hard-to-find glass-transition temperatures and heavily cross-linked polymers [67].

Cross-link density and molecular weight between cross-links were calculated from the DMA data and are presented in Table 2. Results showed that cross-link density and molecular weight between cross-links of the polymers were highly dependent on the functionality of the used acrylate and acetoacetate. If higher functionalized monomers were used, the cross-link density of the obtained polymer was higher, and the molecular weight between cross-links was smaller, which significantly affects the polymer’s mechanical properties.

The highest cross-link density (3.17 × 10^−3^ moles·cm^−3^) and the lowest molecular weight between cross-links (315 g·mol^−1^) were obtained from E^IR^TOFA_TMP_AA and PETA (tetrafunctional acrylate). It can be explained by the higher functionality of both—the Michael donor and the Michael acceptor compound. The lowest cross-link density and molecular weight between cross-links were for the polymers derived from the lower functionalized acetoacetate and difunctional acrylate—BPAEDA.

### 3.5. Tensile Tests

Values of tensile modulus (Young’s) and tensile strength for the samples are shown in Figure 7. Polymer monolith samples obtained by the reaction of polyol acetoacetates with tetraacrylate PETA showed the highest tensile modulus values. The results of tensile strength showed the same trend. This could be explained by the higher functionality of the acrylate and, thus, the higher cross-link density (see Table 2), resulting in the highest module value. A correlation between the functionality of acrylate used in polymer formulations and the tensile module values was observed: the higher functionality of the used acrylate yielded polymers with higher tensile module values.

The tensile strength for NEO380_AA_PETA polymer was 51 MPa. Relatively high modulus values were also obtained for polymers from tall oil from E^IR^TOFA_TMP_AA_TMPTA and E^IR^TOFA_TMP_AA_PETA, respectively, at 2250 MPa and 2370 MPa, and tensile strength was 40 MPa and 44 MPa. Slightly higher average tensile strength and modulus values were for the PETA acrylate, but they were lower for the TMPTA acrylate. These materials can be compared to PMMA, which had a tensile strength value of at least 55 MPa and a tensile modulus of about 2700 MPa [68,69]. Fleischer et al. prepared non-filled NIPU from hexamethylene diamine and glycerol cyclic carbonates: trimethylolpropane. This NIPU showed a tensile strength of 68 MPa and Young’s modulus of 2100 MPa [70]. The mechanical properties of obtained polymers are also comparable to polyurethane materials [71,72]. This shows that tall-oil-based polymer materials obtained by the Michael addition reaction are competitive and promising alternatives.

The highest tensile modulus values were for polymer materials obtained from triacrylate or tetraacrylate and NEO380_AA. A maximum average value of the modulus exceeds 2710 MPa for a NEO380_AA_PETA polymer. The polymers derived using BPAEDA had the lowest modulus value and tensile strength. These polymers (E^IR^TOFA_BD_AA_BPAEDA, NEO380_AA_BPAEDA, and L3300_AA_BPAEDA) exhibited characteristics similar to rubber.

The elongation at break correlates with the modulus value and the tensile strength. The higher modulus and tensile strength values, the smaller the elongation. The smallest elongations at breaks were 1.9%, 2.7%, and 2.9% for polymers E^IR^TOFA_TMP_AA_PETA, NEO380_AA_PETA, and L3300_AA_PETA, and the highest elongations at breaks were 55%, 53% and 47% for polymers E^IR^TOFA_BD_AA_BPAEDA, NEO380_AA_BPAEDA, and L3300_AA_BPAEDA, respectively.

The relationship between mechanical properties and the molecular weight between cross-links was observed. The lower the molecular weight between cross-links, the higher the cross-link density (see Table 2) and the correspondingly higher tensile modulus (Young’s) and tensile strength.

Results indicate that it is possible to obtain very different polymeric materials by changing the chemical structure of the used acrylate. Polymeric materials that are similar in properties to rubbers can be obtained by using lower functionality monomers. It is also possible to obtain materials that are similar in their properties even to organic glass [68,69,73]. By compiling the literature on similar materials and their characteristics, the polymer materials obtained in this study are competitive with several other alternative materials. The production of polymers from fatty-acid-based monomers by Michael reactions is relatively simple and is carried out in milder conditions compared to other methods, such as NIPU synthesis. Michael addition polymerizations can be realized in a short period of time, at room temperature, and at an atmospheric pressure. The chemicals used to synthesize polymers via Michael additions are less toxic than isocyanates, which is one of the main raw materials for polyurethane production. The mechanical and thermal properties of synthesized polymers by the Michael reaction are equal to or even better than materials shown in the literature. Moreover, the properties of polymers can be tailored by varying the monomers used for polymerization. For more information on other similar materials and their characteristics from the literature, see Appendix A Appendix A “Characteristics (glass transition temperatures (T_g_) by DMA, tensile (Young’s) modulus (E^1^), tensile strength and cross-link density (v_e_)) of selected polymers for comparison”.

## 4. Conclusions

Polymer materials were successfully obtained from the synthesized tall oil acetoacetates. The properties of synthesized bio-based polymer monoliths were comparable to polymeric materials obtained from commercially available acrylates. The modulus, tensile strength, and glass transition temperatures were determined for all obtained samples. The properties of the polymer were strongly dependent on the functionality of the monomers used. It was shown that a wide range of polymeric materials with different properties could be obtained by varying the functionality of the monomer. This study demonstrated that polymers can be synthesized with a wide range of mechanical properties, from rubber-like (E^1^ = 8 MPa and σ_max_ = 4 MPa for E^IR^TOFA_BD_AA_BPAEDA) to close to organic glass and NIPUs (E^1^ = 2370 MPa and σ_max_ = 44 MPa for E^IR^TOFA_TMP_AA_PETA). Glass transition temperatures varied from 21.0 °C for E^IR^TOFA_BD_AA_BPAEDA to 63.4 °C for E^IR^TOFA_TMP_AA_PETA obtained by DMA. The synthesized polymers were thermally stable up to 300 °C. Tall-oil-based polymers via the Michael nucleophilic 1,4-addition are an attractive alternative to poly(methyl methacrylate), conventional polyurethanes, NIPUs, and other fossil-based polymer materials. The polymer formulations developed in this study may be suitable for two-component polymer foams, coatings, resins, and composite matrices.

## Figures and Tables

**Figure 1 polymers-14-04068-f001:**
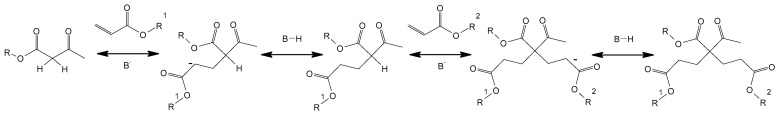
Mechanism of carbon–carbon Michael nucleophilic 1,4-addition reactions.

**Figure 2 polymers-14-04068-f002:**
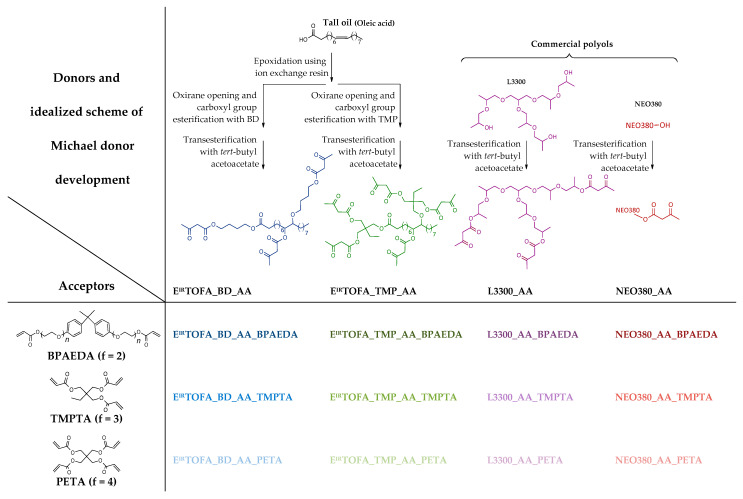
A schematic overview of polymer syntheses and synthesis of Michael donor monomers.

**Figure 3 polymers-14-04068-f003:**
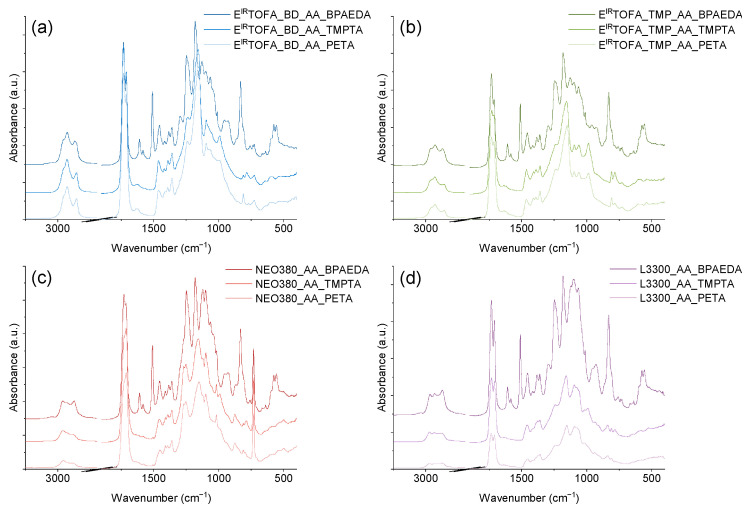
FTIR spectra of polymer monoliths from synthesized acetoacetates and BPAEDA, TMPTA, and PETA acrylates. (**a**) E^IR^TOFA_BD_AA polymers; (**b**) E^IR^TOFA_TMP_AA polymers; (**c**) NEO380_AA polymers; (**d**) L3300_AA polymers.

**Figure 4 polymers-14-04068-f004:**
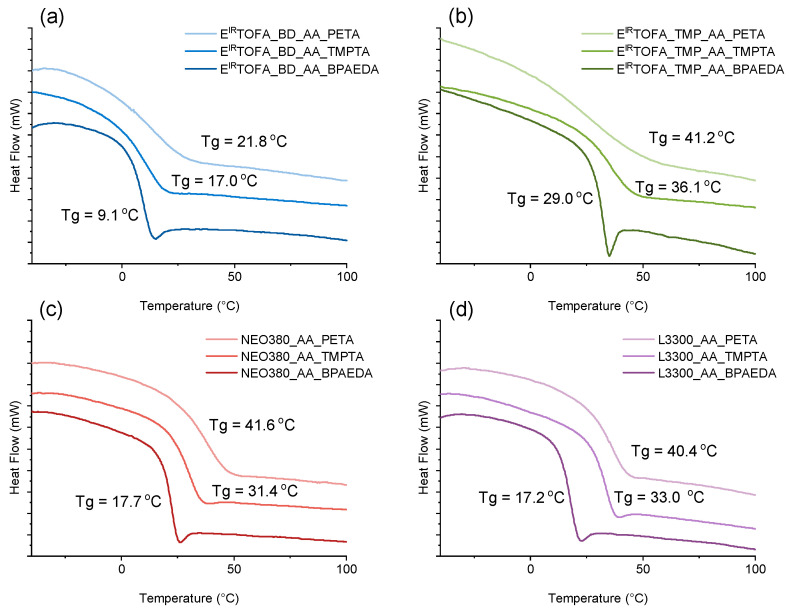
DSC curves of polymer monoliths from synthesized acetoacetates and BPAEDA, TMPTA, and PETA acrylates. (**a**) E^IR^TOFA_BD_AA polymers; (**b**) E^IR^TOFA_TMP_AA polymers; (**c**) NEO380_AA polymers; (**d**) L3300_AA polymers.

**Figure 5 polymers-14-04068-f005:**
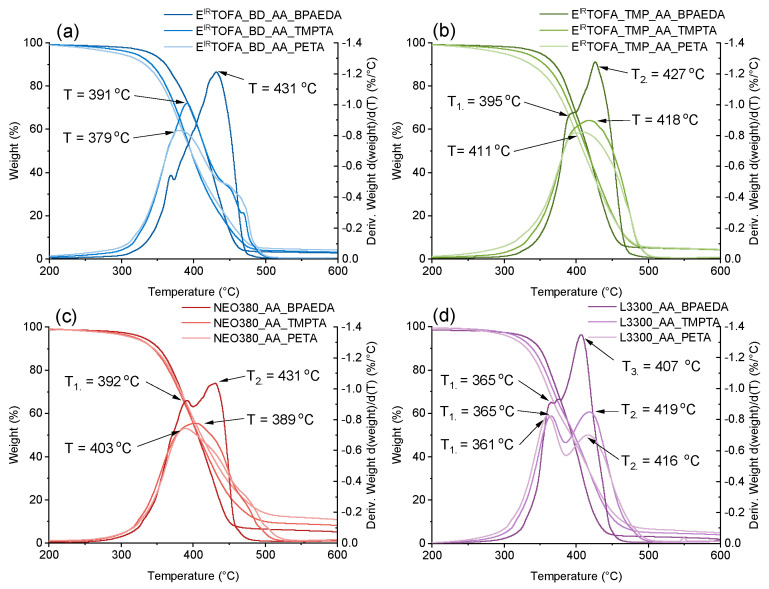
TGA mass loss curve of polymer monoliths from synthesized acetoacetates and BPAEDA, TMPTA, PETA acrylates, and derivative of the mass loss. (**a**) E^IR^TOFA_BD_AA polymers; (**b**) E^IR^TOFA_TMP_AA polymers; (**c**) NEO380_AA polymers; (**d**) L3300_AA polymers.

**Figure 6 polymers-14-04068-f006:**
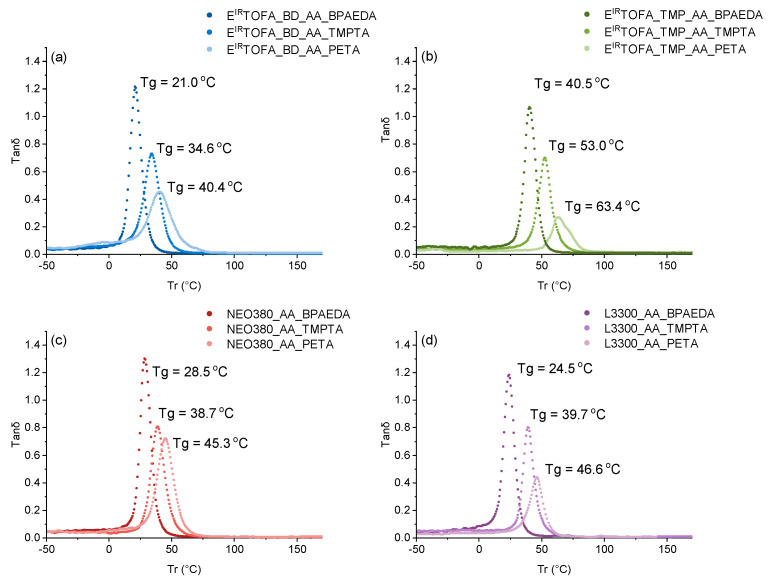
Tanδ curves of polymer monoliths from synthesized acetoacetates and BPAEDA, TMPTA, and PETA acrylates. (**a**) Polymers from E^IR^TOFA_BD_AA; (**b**) polymers from E^IR^TOFA_TMP_AA; (**c**) polymers from NEO380_AA_TMPTA; (**d**) polymers from L3300_AA.

**Figure 7 polymers-14-04068-f007:**
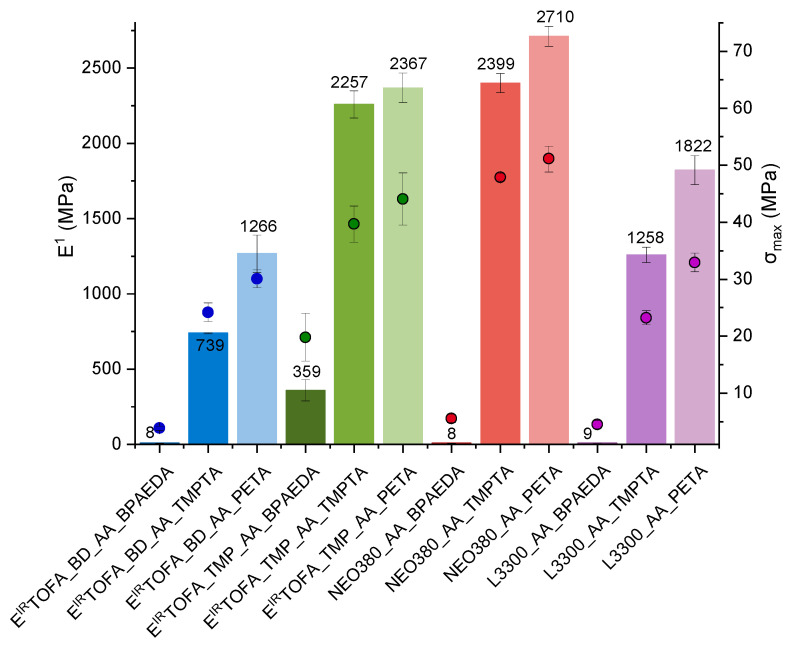
Tensile modulus (Young’s) (E^1^) and tensile strength (σ_max_) for the polymers from tall oil polyol or commercial polyol-based acetoacetates and BPAEDA, TMPTA, and PETA acrylates.

**Table 1 polymers-14-04068-t001:** Weight loss dependence on temperature of synthesized polymers from tall oil polyol or commercial polyol-based acetoacetates and BPAEDA, TMPTA, and PETA acrylates.

	Sample	First Onset, °C	T_m5%,_ °C	T_m10%,_ °C	T_m25%,_ °C	T_m50%,_ °C	Residue, %
Tall oil-based	E^IR^TOFA_BD_AA_BPAEDA	349.0	342.0	361.5	388.1	417.0	2.0
	E^IR^TOFA_BD_AA_TMPTA	342.2	318.3	341.8	369.0	396.2	1.8
	E^IR^TOFA_BD_AA_PETA	340.2	297.2	332.9	365.2	396.1	2.6
	E^IR^TOFA_TMP_AA_BPAEDA	360.9	351.2	370.2	391.8	416.7	3.0
	E^IR^TOFA_TMP_AA_TMPTA	354.1	330.9	356.3	385.8	415.3	2.6
	E^IR^TOFA_TMP_AA_PETA	344.7	300.3	337.7	377.7	410.0	2.8
Commercial polyols-based	L3300_AA_BPAEDA	342.5	331.3	349.2	368.5	394.2	0.1
	L3300_AA_TMPTA	418.8	320.8	339.0	362.8	397.4	2.1
	L3300_AA_PETA	332.3	309.7	331.2	356.6	393.0	2.7
	NEO380_AA_BPAEDA	347.1	331.6	352.7	378.1	406.3	2.1
	NEO380_AA_TMPTA	341.0	321.0	344.5	374.0	407.3	6.0
	NEO380_AA_PETA	347.5	317.0	345.4	376.5	411.3	8.2

**Table 2 polymers-14-04068-t002:** Glass transition temperatures (T_g_), cross-link density (ν_e_) and molecular weight between cross-links (M_c_) from DMA tests and density (ρ) of polymer monoliths from tall oil polyol or commercial polyol-based acetoacetates and BPAEDA, TMPTA, and PETA acrylate.

	Sample	Temperature Range, °C (Tanδ > 0.06 °C)	Max. Tanδ	T_g_, °C	ν_e_, moles·cm^−3^	ρ, g·cm^−3^	M_c,_ g·mol^−1^
Tall oil-based	E^IR^TOFA_BD_AA_BPAEDA	2.9–36.4	1.21	21.0	0.59 × 10^−3^	1.167	1695
	E^IR^TOFA_BD_AA_TMPTA	2.0–50.7	0.73	34.6	1.98 × 10^−3^	1.211	505
	E^IR^TOFA_BD_AA_PETA	−20.0–64.5	0.45	40.4	2.67 × 10^−3^	1.219	375
	E^IR^TOFA_TMP_AA_BPAEDA	21.9–54.5	1.07	40.5	1.89 × 10^−3^	1.116	529
	E^IR^TOFA_TMP_AA_TMPTA	32.6–68.9	0.70	53.0	2.04 × 10^−3^	1.177	490
	E^IR^TOFA_TMP_AA_PETA	52.2–80.2	0.27	63.4	3.17 × 10^−3^	1.235	315
Commercial polyols-based	NEO380_AA_BPAEDA	9.9–43.9	1.30	28.5	0.93 × 10^−3^	1.257	1075
	NEO380_AA_TMPTA	22.1–56.8	0.81	38.7	2.04 × 10^−3^	1.328	490
	NEO380_AA_PETA	20.8–62.7	0.72	45.3	2.55 × 10^−3^	1.330	392
	L3300_AA_BPAEDA	1.0–36.5	1.18	24.5	1.13 × 10^−3^	1.171	885
	L3300_AA_TMPTA	22.5–55.1	0.80	39.7	2.20 × 10^−3^	1.202	455
	L3300_AA_PETA	28.5–57.5	0.44	46.6	2.70 × 10^−3^	1.247	369

## Data Availability

The raw data presented in this study are available on request from the corresponding author.

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
