# Peer review of "Bio-Based Polymer Developments from Tall Oil Fatty Acids by Exploiting Michael Addition"

_polymers, 2022, doi:10.3390/polym14194068_

Round 1
Reviewer 1 Report
1. The potential impact of the research should be highlighted from the beginning of the manuscript. The authors should explain how the results presented in the manuscript will impact the field in relation to other polyurethane advancements.
2. The chemical characterization should include ssNMR besides the FTIR analysis to reveal the chemical composition of the material.
3. What is the intended application of the monoliths? It is unclear why monolith format was chosen, and how it will be used.
4. Reproducibility should be demonstrated. Mass balances should also be closed for the polymerization reactions and compared.
5. Figure and table captions should be longer and more detailed so that the figures and tables can stand on their own without having to read the entire main text. E.g. Figure 1 caption consists of 2 words only.
6. Some errors are reported as standard deviations. However, it is unclear how these were derived and how many independently prepared materials were measured. Elaborate on this more, and give all the details on the sample preparation and repeated experiments.
7. Recent examples of Michael reaction in network polymer synthesis should be acknowledged briefly (10.1002/pol.20210388; 10.1039/D0GC03226C; 10.1021/acs.macromol.0c02128).
8. Both the quotient (“x/y”) and negative exponent (“x y-1”) formats are used in the manuscript for units. Either of them should be used consistently, preferably the negative exponent format, which is recommended by the IUPAC.
9. Comparison to the literature with similar materials and their preparation as well as characteristics should be provided in a table format at the end of the results section.
10. The conclusion section is weak. It is very short and does not have any scientific conclusion but vague discussions. The main research findings should be summarized in quantitative statements.
Reviewer 2 Report
Manuscript title: Bio-based Polymer Development from Tall Oil Fatty Acids by Exploiting Michael Addition 3
Manuscript id: polymers-1919468
Authors: Pomilovskis et al.
The manuscript regarding the topic and results presented is of interest to polymer science community and revisions based on the comments below are recommended before considering for publication.
Major comments
· Insufficient Abstract: In the abstract, the main aim and the significant result of this study are missing, the current version it only highlights the material method. In addition, it would be even better to have a sentence as a future perspective.
· The unit/abbreviation is not mentioned before, consider defining the abbreviation when mentioned for the first time…. Please check throughout the manuscript to define the abbreviations.
· The reference numbers are mixed, consider arranging them in order….
· Line 52-58, the aim or hypothesis of the study is clear, however, the approach is missing ….
· Lake of scientific literature to support the statements and findings throughout the manuscript…... I have made some suggestions for that and more need it….
· More information is needed for ALL TABLE captions and define the abbreviation and units that are used. And adjust the significant figures for the table and manuscript.
· Grammar and punctuation issuers need to be addressed. I have selected/mentioned some as examples.
· I am not sure whether the ‘’…..’’ term is well discussed in the abstract and manuscript. Please consider discussing it or rephrasing it.
· I have a major concern about the results and discussion section. The authors describe the results and compare the results with previous studies, however, insight mechanisms are still insufficient.
· This section is repeating information already presented and explaining things in an unnecessarily complicated way. The quality of the manuscript would benefit from the whole section being condensed.
· The language is generally clear, with some exceptions where the authors are a bit too innovative with the terminology, although there are other good terms to use…..eg.
Minor comments:
Abstract
Line 16-17: A complicated sentence, please revise and check the grammar
Introduction:
Line 30-33: A reference is needed here.
Line 38-40: A complicated sentence, please revise and check the grammar
Line 41-44: A reference is needed here, for example, you can use: https://doi.org/10.1002/adv.21629
Line 53-54 A reference is needed here.
Line 86: Delete ‘’approximate’’, there is no such a thing as approximate structure – did you mean ‘elucidating the structure tentatively’’
In MM section
Literature references are missing for all sub-section. It would be better to cite the references that the procedure adopted.
Additional info is needed for the table caption, most importantly significant figures.
In MM section, what is the quality control (QC) data? There is no mention of the QC.
What is the accuracy of the instruments, recovery, LOD, and LOQ ……. These parameters are needed to report the efficiency of any analytical system.
In general, how many times you’ve recorded the data,? duplicate? Triplicate?..... what you mentioned in the text is not clear, please elaborate more on this
R&D section
These two paragraphs belong to the introduction section, please consider rephrasing or moving the paragraph to the introduction.
These sections are repeating information already presented and explain things in an unnecessarily complicated way. The quality of the manuscript would benefit from the whole section being condensed, Line 219-238 and Line 282-292.
Figure 3. The intensity of the peaks in all panels are not clear – It would be better to change the peaks to black color, they will visualize better.
Figure 7. What the black lines represent- it is not clear. Consider adding the description in the Figure caption.
Line 202: A reference is needed here.
Line 229: A reference is needed here.
Conclusion
I believe there are other a lot of important conclusions that could be made from this study…. And the future perspectives for the following research are highly crucial here.
Round 2
Reviewer 1 Report
The comments were addressed .
Author Response
Thanks to the reviewer for the review. We have made the necessary corrections in the manuscript.
Reviewer 2 Report
I am happy to see the manuscript improved nicely. The authors addressed all my comments adequately.
However to make sure the statements are supported by literature I will recommend adding citations in the following lines:
Line 65:
https://doi.org/10.1039/C9PY01686D
or
https://doi.org/10.1002/pi.5573
Line 204-208:
https://doi.org/10.1002/app.41446
Author Response
Thanks to the reviewer for the suggestions. We have made the necessary corrections in the manuscript according to the suggestions.